# Should Preimplantation Genetic Testing (PGT) Systematically Be Proposed to *BRCA* Pathogenic Variant Carriers?

**DOI:** 10.3390/cancers14235769

**Published:** 2022-11-24

**Authors:** Lucie Laot, Charlotte Sonigo, Julie Nobre, Alexandra Benachi, Traicie Dervin, Lina El Moujahed, Anne Mayeur, Dominique Stoppa-Lyonnet, Julie Steffann, Michael Grynberg

**Affiliations:** 1Department of Reproductive Medicine and Fertility Preservation, Université Paris-Saclay, Assistance Publique Hôpitaux de Paris, Antoine Beclere Hospital, 92140 Clamart, France; 2Inserm, Physiologie et Physiopathologie Endocrinienne, Université Paris Saclay, 94276 Le Kremlin-Bicêtre, France; 3Department of Obstetrics and Gynecology, DMU Santé des Femmes et des Nouveau-nés, Université Paris-Saclay, Assistance Publique Hôpitaux de Paris, Antoine Beclere Hospital, 92140 Clamart, France; 4Service de Biologie de la Reproduction—CECOS, Université Paris-Saclay, Assistance Publique Hôpitaux de Paris, Antoine Beclere Hospital, 92140 Clamart, France; 5Department of Oncology Genetics, Institut Curie, 75005 Paris, France; 6Imagine Institute, INSERM UMR1163, Service de Génétique Moléculaire, Groupe Hospitalier Necker-Enfants Malades, Université de Paris-Sorbonne Paris Cité, AP-HP, 75015 Paris, France

**Keywords:** *BRCA* pathogenic variant, preimplantation genetic testing, fertility preservation

## Abstract

**Simple Summary:**

Preimplantation genetic testing (PGT) has been developed to avoid the transmission of a critical hereditary disease, by selecting embryos for uterine transfer using a genetic analysis. *BRCA* mutated patients can undergo PGT and also be offered fertility preservation (FP) when diagnosed, with cancer or even preventively before a malignancy occurs. However, PGT but also FP success rates are closely related to ovarian reserve parameters, which are known to be potentially lower for *BRCA* pathogenic variant carriers. To this day, although *BRCA* pathogenic variants are related to reproductive issues, there are no international guidelines for the application of PGT and FP in this subgroup of patients. The aim of this article is to review the published real-life data regarding *BRCA* carriers’ ovarian reserve and PGT success rates in oncologic and non-oncologic contexts, to determine the actual indication of PGT and further to improve patients’ care pathway.

**Abstract:**

Over the past years, *BRCA* genes pathogenic variants have been associated to reproductive issues. Indeed, evidence indicate that *BRCA*-mutated patients are not only at higher risk of developing malignancies, but may also present a reduction of the follicular stockpile. Given these characteristics, *BRCA* patients may be candidates to fertility preservation (FP) techniques or preimplantation genetic testing (PGT) to avoid the transmission of this inherited situation. Since the success rates of both procedures are highly related to the number of oocytes that could be recovered after ovarian stimulation, predicted by ovarian reserve tests, they are ideally performed before the diagnosis of cancer and its treatment. Despite the specific reproductive challenges related to *BRCA* status, no international guidelines for the application of PGT and FP in this subgroup of patients is currently available. The present article aims to review the available data regarding *BRCA* carriers’ ovarian reserve and PGT success rates in oncologic and non-oncologic contexts, to determine the actual indication of PGT and further to improve patients’ care pathway.

## 1. Introduction

The discovery of *BRCA1* and *BRCA2* gene pathogenic variants has made it possible to identify high oncologic risk populations and improve their life expectancy with individualized screenings, early cancer diagnosis, novel therapies and prophylactic procedures [1]. These multiple interventions in a woman’s life can impact her quality of life in many areas, such as body image with breast surgery, secondary effects of treatments, sentimental life, professional career or a recurring anxiety due to oncological risk. One of the most important challenges for these women is reproductive outcomes as their childbearing projects can be jeopardized by gonadotoxic treatments or age-related fertility decline in the case of a delayed pregnancy project. Therefore, fertility preservation is recommended before cancer treatments in case of breast cancer diagnosis and is a keystone of the care pathway, but its indications are not restricted to the oncological field only. Indeed, a premature ovarian reserve decrease and a shortening length of reproductive life has been observed among women carrying a *BRCA* pathogenic variant [2,3], making them ‘good’ candidates for fertility preservation procedures, whatever their cancer status. It is recommended to address this issue early to increase patients awareness [4,5]. 

Progress in assisted reproductive medicine has led to the development of a technique called preimplantation genetic testing (PGT), which is designed to avoid the transmission of a critical hereditary disease, by selecting embryos for uterine transfer by means of a genetic analysis [6]. In 2003, in view of an impaired quality of life and a 50% chance of transmission, which could potentially jeopardize a future parental project, the Ethics Taskforce of European Society of Human Reproduction defined as acceptable the extension of PGT to late-onset or incomplete penetrance diseases, such as hereditary predisposition to breast and ovarian cancer due to *BRCA1* or *BRCA2* pathogenic variants [7], and this indication for PGT was also officially approved by the Human Fertilization and Embryology Authority in 2006 [8]. The chances of pregnancy with PGT require a higher oocyte yield than in vitro fertilization (IVF) without genetic analysis, especially in cases of dominant transmission [9]. Thus, a live birth with *BRCA* related PGT usually entails a long and exhaustive process, potentially adding up to previous medical and personal hardships. Success rates are closely related to ovarian reserve parameters, which are known to be potentially lower for *BRCA* pathogenic variant carriers. Therefore, it seems relevant to wonder if *BRCA* related PGT can negatively affect patients’ quality of life in certain situations. By analyzing evidence available in medical literature, the aim of this review article is to discuss the psychological repercussions of PGT given the specific challenges of reproductive medicine in the case of *BRCA* pathogenic variant, in order to determine if this technique should systematically be proposed to *BRCA* pathogenic variant carriers.

## 2. Feasibility of Pre-Implantation Genetic Testing for Monogenic Disorder (PGT) for *BRCA* Pathogenic Variant Carriers

### 2.1. Reasons for Allowing PGT for BRCA Pathogenic Variant Carriers 

PGT was developed in the 1990s in the UK. The aim of this technique is to avoid the transmission of a critical hereditary disease, by analyzing embryos and detecting a pathological variant in a single gene (monogenic PGT or PGT-M), aneuploidies (PGT-A), or chromosomal structural rearrangements (PGT-SR), before embryo intra-uterine transfer. PGT-M was initially allowed for high penetrance diseases, such as autosomic autosomal dominant pathologies (osteogenesis imperfecta for example), recessive dominant or certain X linked diseases (Duchenne muscular dystrophy, Haemophilia, X-fragile syndrome) [6]. This technique provides an alternative to prenatal diagnosis (PND) and avoids termination of pregnancy. In 2003, indications of PGT wereextended to late-onset or incomplete penetrance diseases, such as hereditary predisposition to cancer as *BRCA1* or *BRCA2* mutations [7]. This decision was debated, since a *BRCA* status does not always lead to cancer, can be closely monitored with individualized screening, and is accessible to prophylactic procedures to reduce oncological risk. However, in view of an impaired quality of life and a 50% chance of transmission, which could potentially jeopardize a future parental project, this indication for PGT was officially approved by the Human Fertilization and Embryology Authority in 2006 [8]. Nevertheless, it is to this date still subject to debate [10]. 

Some studies evaluated *BRCA* pathogenic variant carriers’ opinion on the risk of pathogenic variant transmission. In 2008, a study by Staton et al. [11] reported the opinion of 213 women with a *BRCA* pathogenic variant (average age of 34 years old); 88% described an extreme and recurring anxiety regarding the possibility of transmitting their mutation to their offspring. In 2009, 77 patients undergoing *BRCA1/2* testing, awaiting their results, responded to a questionnaire evaluating the potential impact of this mutation on their desire for parenthood [12]; 12% reported that they would not pursue a parental project in the case of a positive result.

In addition to offering relief of a heavy psychological burden regarding childbearing choices, a reason for promoting PGT for *BRCA1/2* could be its cost-effectiveness. Indeed, a study was led by Lipton et al., in 2020, comparing the cost-effectiveness of PGT for *BRCA* pathogenic variant transmission vs. the cost of screenings, prophylactic surgeries and cancer care in the case of *BRCA* transmitting to the next generation [13]. The conclusion of this work was a probability of 98.4% (*BRCA1*) and 97.3% (*BRCA2*) in favor of PGT.

### 2.2. Fundamental Principles of PGT: Feasibility and Chances of Success According to Ovarian Reserve

In France, *BRCA*-related PGT is achievable, but heterogeneous among the territory, as its authorization by Fetal Medecine Centers that are responsible for PGT differs among centers. Certain couples might obtain a positive answer to their request for PGT in one center after having been previously refused by another. Studied criteria to grant an authorization for a PGT request will be the number of cancers in family history, their age onsets, the number of deaths due to *BRCA* related cancer and ages of death.

PGT requires the realization of an IVF with intracytoplasmic sperm injection (ICSI). A biopsy of one or multiple embryo cells will be performed between the 3rd and 5th day of development for a genetic analysis [14,15], and only embryos without a *BRCA* pathogenic variant will be transferred in the patient’s uterus. However, in some cases, no embryo is suitable for intra-uterine transfer, due to an impossibility to perform genetic diagnosis, or if the whole embryo cohort is found to be carrying the pathogenic variant [16]. Given the autosomal dominant nature of *BRCA* transmission, it can be estimated that 50% of the embryos will not be transferred. Nevertheless, it might be discussed to propose to transfer male mutated embryos. If this strategy is accepted, only female mutated embryo will not be transferred representing, theoretically, 25% of the embryos. Therefore, the higher the number of obtained embryos is, the higher the chances will be of obtaining an embryo suitable for transfer. The relationship between the total number of collected oocytes and live birth rates (LBR) has been demonstrated in classical IVF procedures: the evaluation of more than 400,000 cycles of ovarian stimulation for IVF between 1991 and 2008 by Sunkara et al., found a linear relationship until 15 collected oocytes with a maximum LBR, plateauing between 15 and 20, and steadily declining beyond 20 [17]. More recently, a meta-analysis confirmed this data suggesting that the retrieval of 12–18 oocytes is associated with maximal fresh LBR. Nevertheless, a continuing positive association seems to be present between the number of oocytes retrieved and cumulative LBR [18]. This association was also found in PGT contexts by Vandervorst et al., in 1998, with a significantly higher chance of obtaining transferable embryo (s) if more than nine oocytes had been collected [19]. The association between LBR and the number of retrieved oocytes, the number of biopsied embryos and the number of suitable blastocysts to transfer was confirmed later in a study analyzing LBR after PGT among 175 couples (all PGT indications considered) [9]. For the 145 couples undergoing ovarian stimulation for PGT-M, an oocyte yield of 15 eggs was associated with a LBR of 50%. Therefore, it can be concluded that a rich ovarian reserve is an essential prognostic criteria to maximize the chances of PGT success. This point can be problematic for women seeking PGT for *BRCA* pathogenic variant, as ovarian reserve could be altered due to cancer treatment or even in the absence of gonadotoxic treatment.

### 2.3. Ovarian Reserve in Case of BRCA Pathogenic Variant

Numerous parameters have been studied among *BRCA* mutated women in order to assess their ovarian function and ovarian reserve, such as age onset of menopause [20,21,22,23], AMH (anti-Müllerian hormone) levels [2,24,25,26,27,28,29,30,31,32,33,34,35], parity [21,34,36,37,38,39], the rate of assisted reproductive techniques use and infertility [38]. If no difference was found regarding parity and the use of assisted reproductive techniques, several studies evidenced an earlier onset of menopause [20,21,22] and lower levels of AMH [2,25,30,32] when compared with non *BRCA* populations. These conclusions are not contradictory as AMH is a proxy for the quantitative assessment of ovarian reserve but does not reflect fertility. It should be noted that the populations studied were different from one investigation to another (the presence or absence of cancer, fertility preservation or IVF) (Table 1). Even though results were discordant among different studies, this could suggest a shortening of reproductive life by 2 to 3 years and a possible premature diminishing of ovarian reserve in the case of *BRCA* pathogenic variant.

From a pathophysiological point of view, it has been hypothesized that this apparent decline in ovarian reserve could be due to *BRCA1* and *BRCA2* involvement in the homologous recombination pathway: an anomaly of this double strand genetic repair system may lead to an anticipated apoptosis of oocytes [3,40,41]. Other genes belonging to this pathway have been identified as predisposing to premature ovarian insufficiency, such as *ATM* (ataxia telangiectasia mutated) bi-allelic mutation [42]. An in vitro analysis on mice oocytes supports this hypothesis, by describing an accumulation of double strand breaks in primordial follicles with age [2]. In a *BRCA1* mutated mice model, a decrease of primordial follicle count was observed, as well as an increase of double strand breaks rate in the remaining follicles. Finally, *BRCA1*/2 could also be implied in telomeres’ length maintenance, which has been associated with reproductive aging [43].

As *BRCA* pathogenic variant carriers are not only concerned by a risk of breast and ovarian cancer, but also by a risk of premature ovarian aging, they could be eligible for fertility preservation procedures, even in non oncologic contexts, with oocyte and/or embryo cryopreservation, after ovarian stimulation [4]. Thus, ovarian stimulation could be performed for *BRCA*-mutated women in several situations: fertility preservation in oncologic or not oncologic context, or for PGT.

### 2.4. Response to Ovarian Stimulation in Case of BRCA Pathogenic Variant

Multiple research works have tried to determine if *BRCA* mutation could impact patients’ response to stimulation and therefore lower their chances of success with assisted reproductive techniques. A study by Oktay et al., in 2010, was the first to evidence a diminished response to ovarian stimulation in the case of *BRCA1* mutation in breast cancer (BC) fertility preservation contexts [44]. These results were later confirmed by several studies in IVF/PGT cycles for *BRCA1* and *BRCA2* patients and in BC fertility preservation contexts [32,45,46,47]. Nevertheless, contradictory results were found in other recent studies for BC fertility preservation and IVF, and for oncologic and non oncologic fertility preservation [27,29,32,48]. These results are summarized in Table 2.

Despite discordant results, this data suggests that *BRCA* pathogenic variant could entail a diminished response to ovarian stimulation. It should be noted that these results were not always adjusted on the presence of cancer, which has recently been hypothesized as an independent cause of poor response to ovarian stimulation [49].

### 2.5. Safety of Pregnancy and Hormonal Stimulation for BRCA Carriers

In the last decade, it has been extensively demonstrated in the general population that pregnancy diminishes the global risk of BC, and that pregnancy after BC does not constitute a risk factor of relapse [50]. Less data is available regarding the specific situation of *BRCA* carriers. One study was published in 1999, stating that BC risk increased with parity when studying 472 *BRCA* patients [51]. These results were invalidated by two ulterior studies of 1601 and 789 patients, describing a protective effect of pregnancy [52,53]. The impact of pregnancy after BC was studied in a multicentric cohort of *BRCA* patients led by the Hereditary Breast Cancer Clinical Study Group. When comparing 128 patients having had a pregnancy during or after BC treatments with 269 patients who did not have a pregnancy after their cancer, no difference was found in terms of overall survival [54]. The same conclusion was reached by the study of 1252 *BRCA* patients with a BC diagnosis, 195 of whom had a pregnancy after BC without any impact on relapse free survival and overall survival after a median follow up of 8.3 years [55].

The impact of hormonal stimulation treatments was also studied in the case of *BRCA* pathogenic variant, but almost exclusively in cancer free situations. No increase of BC risk was found in a population of 1380 women in the case of previous IVF, and this remained true after adjusting on parity [56]. These results were confirmed later in a study including 2514 patients [57]. The risk of ovarian cancer was also ruled out whatever the type of treatment used for 175 patients in a cohort of 1073 women: clomiphene citrate, gonadotropins, or both [58]. The same results were obtained by Gronwald et al., in 2016, in a case control study of 941 pairs of *BRCA* patients [59]. This data is reassuring, stating the absence of over risk of cancer for *BRCA* carriers wishing to undergo hormonal stimulation for fertility preservation, or assisted reproductive techniques in non-oncological contexts.

Data regarding the safety of hormonal stimulation in the case of BC for *BRCA* carriers also seems reassuring, despite being far more limited. Kim et al. compared the oncologic prognosis of 26 *BRCA* mutation carriers with a BC diagnosis, having undergone hormonal stimulation for fertility preservation with 26 other patients without fertility preservation procedures. No difference was found regarding overall survival. More studies are necessary to establish the safety of ovarian stimulation for *BRCA* patients in oncologic contexts, whether it is for fertility preservation before treatments, or for assisted reproductive techniques after treatments [60].

### 2.6. Implications for Reproductive Outcomes of BRCA-PGT

The quoted above medical literature has shown that an oocyte yield of nine or less is a common situation for *BRCA* pathogenic variant carriers (cf. Table 2), but also constitutes a poor prognosis factor for LBR after PGT techniques [9]. A way to overcome this limit could be an oocyte accumulation strategy by realizing multiple cycles of ovarian stimulation. Since oocyte yield is mostly conditioned by age, the ideal situation would be to discuss the topic of *BRCA* related PGT in cancer free contexts, and as early as possible, for example before the age of 30, as it was suggested in a recent review [4].

However, in an urgent situation for oncologic fertility preservation, oocyte accumulation is usually impossible, especially in the case of neoadjuvant chemotherapy, and the number of vitrified oocytes might be too low to be able to obtain a suitable blastocyst for transfer after PGT. Thus, LBR with *BRCA*-PGT could be drastically diminished. Multiple ovarian stimulation cycles can sometimes be authorized after treatments, but this can occur several years after BC diagnosis, resulting in an altered response to hormonal stimulation or in some cases premature ovarian insufficiencies due to age and/or previous gonadotoxic treatments. In these situations, and whatever the number of vitrified oocytes, the chances of obtaining an embryo suitable for intra uterine transfer will most likely be greater with classical IVF than with *BRCA*-PGT.

It seems natural to deliver the most complete information to patients, mentioning the possibility of PGT while including the length and heaviness of treatments, the low chances of pregnancy, live birth, and the risks of unachievable PGT diagnosis. This can add a psychological burden to patients who have already had a long and exhaustive medical history, while resulting in a negative reproductive outcome at the expense of patients’ well-being. Therefore, it seems legitimate to question the feasibility and profitability of PGT for *BRCA* carriers, especially in oncologic contexts. At the present time, there are no international guidelines regarding PGT in *BRCA* mutation contexts to guide medical professionals towards a systematic information of PGT or towards a case-by-case attitude.

## 3. PGT for *BRCA1*/2 Pathogenic Variants: Literature Review

### 3.1. Healthcare Providers and Patients’ Levels of Awareness

The first live birth due to *BRCA* related PGT occurred in 2008 in a context of *BRCA1* mutation [61]. Since then, *BRCA*-PGT has developed, but patients’ awareness remains low. Several studies have highlighted a lack of knowledge from patients and from healthcare providers. In 2009, it was estimated that 32% of *BRCA* carriers knew about the existence of PGT [62], and this percentage has since then increased to approximately 66% in a study published in 2017, interrogating 191 patients [63]. A recent study led by Gietel-Habets et al. in the Netherlands (where hereditary predisposition to breast and ovarian cancer is one of the most frequent indications for PGT) evaluated the level of awareness of 188 medical professionals working in the fields of breast cancer, ovarian cancer or reproduction [64]. Among them, half knew about the existence of PGT in *BRCA* contexts, and the majority had little to moderate knowledge about it. In total, 86% considered it as an acceptable proposal, and 48% had already addressed *BRCA* patients in PGT consultations.

### 3.2. Modalities of BRCA-PGT Announcement: Acceptance and Psychological Impact on Patients

The PGT acceptance rate among *BRCA* patients has been evaluated by multiple research works over the last 10 years [11,12,62,63,65,66,67,68,69,70,71,72]. Most of them report a high percentage of patients viewing PGT as an acceptable option for *BRCA* carriers, but a lower percentage would actually consider it for themselves. The main studies assessing *BRCA*-PGT acceptance are summarized in Table 3.

Overall, PGT for *BRCA* pathogenic variant seems better accepted than PND followed by pregnancy interruption [73]. The low percentage of patients willing to undergo PGT for a *BRCA* pathogenic variant seems to be related to multiple factors: a lack of knowledge among physicians and *BRCA* carriers; the lack of availability; the potential cost of the procedure; concerns regarding the efficacy of the procedures; concerns regarding the safety of ovarian stimulation; and pregnancy after breast cancer [15].

### 3.3. Live Births after BRCA-PGT 

Since the first live birth obtained in 2008 with this technique, several *BRCA*-PGT cohorts have been described. In 2009, Sagi et al. [76] described 10 patients with a *BRCA1* or *BRCA2* mutation, aged 29 to 38 years old, who had a PGT consult: three of them had a previous breast cancer, but none were in the process of reusing cryopreserved material, and eight of them were already considering IVF for previous infertility. Overall, six accepted a *BRCA*-PGT strategy: five were cancer free and one had had a BC. A live birth was observed for three couples after their first attempt. The main reason of refusal for the four remaining couples was the decreased chance of LBR compared with classical IVF.

In 2014, Derks-Smeets et al. [77] published an observational study including 70 couples undergoing *BRCA* related PGT: 64 were cancer free, six women had a previous BC, and *BRCA* mutation was carried by the woman in 60% of cases. The median age of the women was 29.5 years old, and 86% of the couples had never procreated. A total of 145 stimulation cycles were studied (2.5 per couple on average) with 720 embryos in total: 40.8% did not carry the mutation, 43.2% had a *BRCA* mutation, 9.7% were considered abnormal and 6.3% could not be diagnosed. Two *BRCA1* patients had a BC diagnosis after hormonal stimulation. Pregnancy rates per transfer were 39.1% for fresh embryo transfer and 26.5% per frozen embryo transfer. These favorable results can be explained with the women’s median age, which was low in this cohort.

On the contrary, in 2017, Dagan et al. [78] described a rate of PGT interruption of 75% in a cohort of 18 couples undergoing *BRCA*-PGT. Women were carrying the mutation in 14 couples, eight of them were cancer free, and they had all experienced previous infertility. All had at least one cycle of ovarian stimulation. Out of the 12 couples undergoing PGT with no history of cancer, only three of them had a live birth with PGT. The nine remaining couples abandoned PGT for medical or personal reasons: five pregnancies occurred with classical IVF afterwards and four did not procreate. When asked about this decision, the main reason for interrupting PGT attempts was emotional burden, intensified by technical, logistical and financial difficulties.

Less data is available regarding LBR with *BRCA*-PGT in oncologic contexts. In the Derks-Smeets et al. study [77], six patients had a BC history and two of them conceived. Two patients had not had fertility preservation and were approximately 35 years old. One of them had a live birth with PGT, and had not had chemotherapy, contrary to the other one. Out of the four patients with cryopreserved material, one chose not to use it and was shortly diagnosed with a BC relapse. The three remaining patients, respectively, had an extra uterine pregnancy, a live birth and no pregnancy. These results were in favor of a systematic use of cryopreserved material as ovarian stimulation will likely not induce a sufficient response due to age related ovarian reserve decline or post chemotherapy premature ovarian insufficiency, as was the case for two patients. In the Dagan et al. study [78], six women had a medical history of BC. Out of the four patients with cryopreserved material, one had a pregnancy with PGT and surrogate, ending in a live birth. The three others did not have a live birth and did not have more assisted reproductive techniques. One had two natural pregnancies afterwards. One of the two patients with no fertility preservation had a live birth with *BRCA*-PGT. In total, medical literature describes 12 *BRCA* carriers with a history of BC having undergone PGT: two patients out of eight with cryopreserved material had a live birth (one with a surrogate), and two patients out of four without fertility preservation had a live birth.

Another essential point to consider is the psychological impact of PGT. This specific theme has been discussed in focus groups with *BRCA* carriers in genetic consults, as well as with couples consulting for assisted reproductive techniques with *BRCA*-PGT. The aim of those studies was to identify the different levels of psychological repercussion induced by the announcement of PGT’s existence, in order to improve the way this information is delivered to patients. In 2010, Quinn et al. reported 13 patients’ interviews [74]. The study population included cancer free patients and patients previously treated for BC. A large majority had been deceived by the quality of oncofertility information they had been delivered, judging it to be insufficient, and wished for a standardization of practices with the mention of *BRCA* related PGT. Besides, patients also described the sensation of being rushed into making a rapid decision about an important topic, which had not been previously discussed with their partner. Some recognized that their *BRCA* status had an impact on their relationship with their partner and their family. Recurring themes were a sense of guilt regarding childbearing desires despite a risk of mutation transmitting, a feeling of responsibility towards a future family and the urgency to start a parental project. Certain participants felt less anxiety in the case of male offspring. Even though they did not always wish to undergo PGT, some patients felt required to consider this technique or even to choose it due to their hereditary risk, and could feel guilty if they did not. Moreover, reserves were expressed regarding embryo selection due to ethical and/or religious beliefs. The study population was unanimous as regards to the necessity of a psychological support from health care providers following PGT announcement.

In a second study, which was published in 2012, 29 patients awaiting *BRCA* genetic mutation results were shown a tutorial about PGT [75]. They all agreed that PGT’s existence should be mentioned during oncogenetic consult, whether patients considered it acceptable or not. However, despite wanting to have the most complete information, they felt overwhelmed by the quantity of information given, especially if receiving a diagnosis of *BRCA* pathogenic variant at the same time. It was hypothesized that a preliminary information could be delivered during a first consult with a pamphlet, and that a second consult could then be dedicated to PGT. Regarding the type of health care provider delivering the information, it seemed obvious that first information could be from a genetic counselor. Given the intimate nature of the issues raised by PGT, such as childbearing, most of them wished to be able to discuss it afterwards with a known professional or a gynecologist. A majority of the study population did not consider it relevant to be briefed about PGT before genetic results, but a few viewed it as a helpful way to apprehend a positive test result by knowing that transmission could be avoided without impacting their childbearing desires. Finally, some patients preferred to differ discussions about PGT and primarily focus on oncologic risk management before discussing reproductive issues because they did not feel concerned yet.

In total, delivering information about *BRCA*-PGT is a complicated step in *BRCA* carriers’ health care pathway, already dealing with increased surveillance and potential prophylactic procedures, while suggesting invasive medical IVF procedures for PGT despite a potential absence of infertility. Information about PGT should be delivered with care, as it can raise issues and dilemma on a personal and ethical level.

## 4. Conclusions

Available, although scarce, medical literature about *BRCA* related PGT suggests that live birth chances are increased when ovarian stimulation is realized before the age of 30 with the possibility of oocyte accumulation. Success rates seem to be lower in oncologic contexts, with the risk of an altered quality of life due to a heavy psychological burden, induced by choosing not to transmit a serious hereditary predisposition at the potential expense of childbearing, and new heavy treatment protocols following previous cancer treatments. Despite the difficulty to draw conclusions from these studies, which include a limited number of patients, it seems that *BRCA* related PGT might not be beneficial for some patients, which raises the question of whether or not PGT should be proposed to all *BRCA* carriers. Information about PGT should be delivered with care, as it can raise issues on a personal and ethical level, and psychological support should be available for patients. To this day, there are no international guidelines regarding PGT in *BRCA* pathogenic variant contexts to guide medical professionals towards a systematic proposal of PGT, or towards a case-by-case attitude depending on ovarian reserve parameters or the existence of cryopreserved oocytes/embryos.

## Figures and Tables

**Table 1 cancers-14-05769-t001:** Studies assessing anti-Müllerian hormone levels of *BRCA* pathogenic variant carriers.

Study	Design	*BRCA* Group (*n*, Average Age)	Control Group (*n*, Average Age)	AMH for *BRCA* Group (ng/mL)	AMH for Control Group (ng/mL)	*p*
Titus et al., (2013) [2]	Cross-sectional	BC (*n* = 24, 34.8)	BC (*n*= 60, 36.3)	1.22	2.23	<0.0001
Michaelson-C et al., (2014) [24]	Cross-sectional	ø cancer (*n* = 41, 33.2)	ø cancer (*n* = 324, NA)	2.71	2.02	0.27
Wang et al., (2014) [25]	Cross-sectional	ø cancer (*n* = 89, 35.5)	ø cancer (*n* = 54, 35.6)	0.53 (*BRCA1*) 0.73 (*BRCA2*)	1.05	0.026 (*BRCA1*) 0.634 (*BRCA2*)
van Tilborg et al., (2016) [26]	Cross-sectional	ø cancer (*n* = 124, 29)	ø cancer (*n* = 131, 31)	1.9	1.8	0.34
Lambertini et al., (2018) [27]	Retrospective	BC (*n* = 29, 31)	BC (*n*= 72, 30)	1.8	2.6	0.109
Grynberg et al., (2019) [28]	Retrospective	BC (*n* = 52, 31.7)	BC (*n*= 277, 32.3)	3.6	4.1	0.3
Gunnala et al., (2019) [29]	Retrospective	BC (*n* = 38) ø cancer (*n* = 19) 32.4	BC (*n* = 53) OM (*n* = 85) EEF (*n* = 600) 35.5	2.8 (overall) 2.4 (*BRCA1*) 3.6 (*BRCA2*) 2.6 (BC) 3.2 (ø cancer)	2.4 (overall) 2.4 (BC) 2.9 (OM) 2.3 (EEF)	0.22 0.6
Son et al., (2019) [30]	Retrospective	BC (*n* = 52, 34.0)	BC (*n* = 264, 34.0)	2.6	3.85	0.004
Lambertini et al., (2019) [35]	Prospective	BC (*n* = 35, 34.0)	BC (*n* = 113, 36.0)	2.82	2.46	0.53
Ponce et al., (2020) [31]	Prospective	ø cancer (*n* = 69, 32.3)	ø cancer (*n* = 66, 32.7)	3 (*BRCA1*) 2.54 (*BRCA2*)	2.27	0.28
Porcu et al., (2020) [32]	Prospective	BC (*n* = 11 *BRCA1*, 31.5) (*n* = 11 *BRCA2*, 33.2)	BC (*n* = 24, 32.5) ø cancer (*n* = 181, 32.4)	1.2 (*BRCA1*) 4.4 (*BRCA2*)	4.5 (BC) 3.8 (ø cancer)	≤0.05

Abbreviations: BC = breast cancer; OM = other malignancy; EEF = elective egg freezing.

**Table 2 cancers-14-05769-t002:** Studies assessing response to ovarian stimulation in case of *BRCA* mutation.

Study	Design	*BRCA* Group (*n*)	Control Group (*n*, Average Age)	Number of Collected Oocytes for *BRCA* Group *	Number of Collected Oocytes for Control Group *	*p*
Oktay et al., (2010) [44]	Prospective	FP for BC (*n* = 8 *BRCA1; n =* 4 *BRCA2)*	FP for BC (*n* = 68) (33 negative, 35 untested)	7.9 (overall) 95% CI [4.6–13.8] 7.4 (*BRCA1*) 95% CI [3.1–17.7]	11.3 *(BRCA* neg) 95% CI [9.1–14.1] 12.4 *(BRCA* neg + ø) 95% CI [10.8–14.2]	0.025 0.03
Shapira et al., (2015) [48]	Retrospective	FP for BC (*n* = 21) no cancer, IVF with PGT (*n* = 41)	FP for BC (*n* = 21) IVF with PGT non *BRCA* (*n* = 41)	13.75 ± 7.6	14.75 ± 8.8	0.49
Derks et al., (2017) [45]	Retrospective	no cancer, IVF with PGT (*n* = 18 *BRCA1; n =* 20 *BRCA2*)	IVF with PGT non *BRCA* (*n* = 154)	7.0 [IQR 4–9] (overall) 6.5 [IQR 4–8] *(BRCA1)* 7.5 [IQR 5.5–9] *(BRCA2)*	8.0 [IQR 6–11]	0.02 0.01 0.2
Lambertini et al., (2018) [27]	Retrospective	FP for BC *n* = 10	FP for BC *n* = 19	6.5 [IQR 3–7]	9.0 [IQR 5–13]	0.145
Turan et al., (2018) [46]	Prospective	FP for BC *n* = 21	FP for BC *n* = 97	7.4 ± 5.7	10.6 ± 5.1	0.047
Gunnala et al., (2019) [29]	Retrospective	FP for BC (*n* = 38) no cancer (*n* = 19)	FP for BC (*n* = 53) OM (*n* = 85) EEF (*n* = 600)	14.0 ± 7.9 (overall) 13.5 ± 7.3 *(BRCA1)* 14.2 ± 6.1 *(BRCA2)* 14.4 ± 9.1 (BC) 13.2 ± 4.7 (ø cancer)	10.4 ± 6.9 (overall) 13.1 ± 8.4 (BC) 14.2 ± 8.9 (OM) 9.6 ± 6.2 (EEF)	>0.05
Porcu et al., (2020) [32]	Prospective	FP for BC (*n* = 11 *BRCA1*); (*n* = 11 *BRCA2*)	FP for BC (*n* = 24) ø cancer (*n* = 181)	6.7 ± 4.9 *(BRCA1)* 10 ± 6.8 *(BRCA2)*	9.1 ± 6.1 (BC) 8.8 ± 4.3 (no cancer)	>0.05
Kim et al. (2022) [47]	Retrospective	FP for BC (*n* = 25 *BRCA1*; *n* = 35 *BRCA2*; *n* = 21 *BRCA* 1 + 2)	FP for BC *n* = 57	8.3 ± 5.4 *(BRCA 1- 2)*	15.3 ± 8.7 (BC)	0 .002

* Mean ± SD or Median 95% CI or Median [IQR]. Abbreviations: BC = breast cancer; OM = other malignancy; EEF = elective egg freezing.

**Table 3 cancers-14-05769-t003:** Rates of *BRCA* related PGT acceptance rates in medical literature.

Study	*n*	Population	Acceptance Rate for other *BRCA* Carriers	Acceptance Rate for Oneself
Menon et al. (2007) [65]	52	*BRCA* mutation carriers attending a Familial Cancer Clinic, with and without personal cancer history	75%	37.5% (retrospectively) 14%
Staton et al. (2008) [11]	213	*BRCA* mutation carriers with and without personal cancer history (online questionnaire)	75%	13%
Vadaparampil et al. (2009) [66]	962	Members of a national organization dedicated to empowering women at high risk for developing breast or ovarian cancer (web-based survey)	NA	33%
Fortuny et al. (2009) [12]	77	Individuals undergoing *BRCA1/2* testing	61% (74% if cancer history, 44% if cancer free)	48% (61% if cancer history, 30% if cancer free)
Quinn et al. (2009) [62]	111	Attendees of a national conference for individuals and families affected by hereditary breast and ovarian cancer	57%	NA
Dekeuwer et al. (2011) [72]	20	20 *BRCA* mutation carriers (13 of childbearing age; BC or ovarian cancer)	NA	35%
Reynier et al. (2012) [67]	605	Unaffected *BRCA1/2* mutation carriers of childbearing age, included at least 1 year after the disclosure of their test results	85%	32.5%
Woodson et al. (2014) [68]	148 114	Awaiting for genetic result (BC family history) After genetic results (Family history of BC)	NA	28% 24%
Pellegrini et al. (2014) [69]	20	*BRCA* mutation carriers, 31 to 57 years old, including 12 with a history of breast/ovarian cancer	NA	70%
Chan et al. (2016) [70]	1081	*BRCA* mutated women with a history of breast/ovarian cancer for 36%	59%	35%
Gietel et al. (2017) [63]	191	*BRCA* mutation carriers	80%	39%
Mor et al. (2018) [71]	70	Married Israeli and healthy *BRCA* mutated women who wanted children before and after receiving BRCA test results	NA	25,7%

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
