# Peer review of "Should Preimplantation Genetic Testing (PGT) Systematically Be Proposed to BRCA Pathogenic Variant Carriers?"

_cancers, 2022, doi:10.3390/cancers14235769_

Round 1

Reviewer 1 Report

Since 2021, the genetic counseling society has retired the term BRCA "mutation" and changed it to the more neutral term " pathogenic or likely pathogenic variant" to describe a hereditary genetic predisposition to cancer. The attitude should also be carried on in the PGT field as an effort to reduce raising possible ethical and psychological issues. Therefore, I suggest the author do so in this review article to demonstrate your professionalism not only in fertility medicine but also in human genetics.

Reviewer 2 Report

This review examines data regarding the usefulness of pre-implantation genetic testing (PGT) among BRCA carriers in order to contribute towards international guidelines regarding the management of fertility preservation in this population.

Overall, the article is well-written and clear. Few comments for improvement:

line 42: edit text ".. aims review to the available..." to "....aims to review available..."

line 78: edit "real life data" to "evidence"

line 86: edit "England" to "UK"

line 132 - 133: (25% if the couple accepts a transfer of male mutated embryos, as it can sometimes be proposed in some countries). The authors should explain why 25% and not 50% if the carrier is male, and provide a reference regarding the countries that this is proposed.

Throughout the manuscript: please avoid the acronym ART and AMH - better to refer to "assisted reproductive techniques" and "anti-Muellerian hormone" every time.

lines 235 and 239: please italicize BRCA

line 256: edit "legit" to "legitimate"

line 260: edit "real life data" to "literature review" or "scientific evidence" or something similar

line 336: please explain "...(including one with sex diagnosis)" What is a sex diagnosis?
